# Current Dengue Virus Vaccine Developments and Future Directions

**DOI:** 10.3390/v17020212

**Published:** 2025-01-31

**Authors:** Govindaraj Anumanthan, Bikash Sahay, Ayalew Mergia

**Affiliations:** Department of Infectious Diseases and Immunology, University of Florida, Gainesville, FL 32611, USAsahayb@ufl.edu (B.S.)

**Keywords:** dengue vaccine, immunogenicity, dengue fever, vaccine efficacy, vaccine safety

## Abstract

Dengue fever (DF), a leading arboviral disease globally, is caused by the Dengue virus (DENV) and represents a significant public health concern, with an estimated 390 million cases reported annually. Due to the complexity of the various dengue variants and the severity of the disease, vaccination emerges as the essential strategy for combating this widespread infectious disease. The absence of specific antiviral medications underscores the critical need for developing a Dengue vaccine. This review aims to present the current status and future prospects of Dengue vaccine development. Further, this review elaborates on the various strategies employed in vaccine development, including attenuated, inactivated, subunit, and viral vector vaccines. Each approach is evaluated based on its immunogenicity, safety, and efficacy, drawing on data from preclinical and clinical studies to highlight the strengths and limitations of each candidate vaccine. The current study sheds light on future directions and research priorities in developing Dengue vaccines. In conclusion, the development of a Dengue vaccine holds significant potential for reducing the global burden of DF. However, challenges remain in terms of vaccine safety, efficacy, delivery, and availability. Overcoming these challenges, coupled with advancements in vaccine technology, could lead to better control and prevention of Dengue, thereby enhancing public health and quality of life.

## 1. Introduction

Dengue is a mosquito-borne viral disease transmitted through the bite of infected mosquitoes, specifically those belonging to the Aedes species, which include Aedes aegypti and Aedes albopictus. These mosquitoes are particularly prevalent in urban settings, especially in tropical and subtropical areas where the disease is endemic [1]. The infection can result in a wide range of clinical presentations, from mild febrile illness to more severe conditions such as dengue hemorrhagic fever (DHF) and dengue shock syndrome (DSS), which can be fatal [2]. Over the past fifty years, the global incidence of dengue has seen a significant increase, with estimates indicating that between 284 and 528 million people are infected each year, with approximately 96 million of these cases showing symptoms [3]. The World Health Organization (WHO) has classified dengue as one of the most important mosquito-borne viral diseases, affecting over half of the global population, particularly in areas at risk of infection [4]. The search for a dengue vaccine has been an extensive and complex endeavor, with researchers exploring various approaches and strategies. Among the most notable advancements in recent years is the introduction of the first licensed dengue vaccine, Dengvaxia, by the French pharmaceutical company Sanofi Pasteur [5]. This vaccine, approved for use in several countries, including Mexico, the Philippines, and Brazil, has been hailed as a significant breakthrough in the battle against dengue [6]. However, the vaccine has also faced challenges, particularly concerns regarding its effectiveness and safety, especially in individuals who have not been previously exposed to the dengue virus [7]. Nonetheless, researchers have pursued alternative avenues for dengue vaccine development, including live-attenuated, inactivated, and subunit vaccines [8,9]. Live-attenuated vaccines, which utilize weakened forms of the dengue virus, have demonstrated promising outcomes in clinical trials, showing the capacity to stimulate a strong immune response in vaccinated individuals [10]. The development of an effective dengue vaccine has been a critical priority for researchers. In this review, we will explore the current landscape of dengue vaccine research, examining the advancements achieved, the challenges encountered, and the potential future directions in this crucial area of study.

### 1.1. The Genome of the DENV Virus

The genome of the dengue virus (DENV) is approximately 11 kilobases long and consists of a positive sense single-stranded RNA (ssRNA) [11]. It features a single, extensive open reading frame (ORF) flanked by two untranslated regions [12], namely the 5′-UTR and 3′-UTR. The 5′-UTR measures approximately 95–101 nucleotides in length [13] and encompasses several significant elements, including SLA, SLB, 5′UAR, 5′DAR, cHP, and 5′CS [14]. Stem-loop A (SLA) functions as a promoter for the viral RNA-dependent RNA polymerase (NS5). Stem-loop B (SLB) contains essential sequences for long-range RNA–RNA interaction and genome replication. The 5′-upstream AUG region (5′-UAR) and the 5′-downstream AUG region (5′-DAR) are involved in genome cyclization. The C-coding region hairpin (cHP), responsible for translation initiation from the C-start codon and 5′-cyclization sequence (5′-CS), mediates RNA–RNA interaction between the 5′- and 3′-ends of the viral genome, which are essential for genome cyclization. The ORF encodes both structural and non-structural proteins, which are translated into a polyprotein and processed co- and post-translationally by cellular and viral proteases to produce ten mature viral proteins. The structural proteins include the capsid (C), pre-membrane/membrane (prM/M), and envelope (E) proteins, while the non-structural proteins include NS1, NS2A, NS2B, NS3, NS4A, NS4B, and NS5 [15,16]. These proteins are involved in various aspects of RNA replication and viral assembly, as shown in Figure 1.

### 1.2. The Structure and Variation of DENV

The structure of the dengue virus (DENV) is characterized by a mature virion that has a smooth surface, approximately 50 nanometers (nm) in diameter [17], while the immature virion measures about 60 nm and has a spiky surface. The DENV genome encodes three structural proteins: the capsid (C), pre-membrane/membrane (prM/M), and envelope (E) proteins, along with seven non-structural (NS) proteins involved in RNA replication [18]. The C protein is crucial for nucleocapsid formation during the early stages of DENV virion assembly, and the M protein plays a significant role in the arrangement and maturation of the DENV particle [19]. The E protein, which consists of three domains, is essential for virus binding and fusion to host cells, with domain III specifically responsible for receptor-binding activity [20]. DENV can be classified into four distinct serotypes (DENV-1, DENV-2, DENV-3, and DENV-4), which share approximately 65% amino acid sequence similarity [21]. Each serotype includes several genotypes, defined as groups of DENV isolates with no more than 6% nucleotide sequence divergence [22]. For instance, DENV-1 can be divided into five genotypes, while DENV-2 has six, and DENV-3 and DENV-4 have four genotypes each [23]. Different serotypes and genotypes may induce varied immune responses and differ in their ability to infect various target cells, which can influence the severity of dengue disease [24]. However, it is apparent that all four serotypes can cause potentially lethal dengue fever (DF), dengue hemorrhagic fever (DHF), and dengue shock syndrome (DSS). Therefore, the DENV vaccine must prevent infection with each of the four serotypes.

### 1.3. Development of Dengue Vaccine

Dengue vaccine development has been a complex and challenging process due to the unique properties of the DENV and the need to provide protection against all four serotypes. In the early stages of dengue vaccine development, whole inactivated or live attenuated viruses were used. However, these early candidates faced challenges regarding safety, immunogenicity, and the ability to elicit a well-balanced immune response against each of the four serotypes. All four DENV serotypes (DENV 1–4) can cause DF, DHF, and DSS. An exclusive feature of DF is that an initial infection (primary infection) with any DENV serotype can provide homotypic and enduring protection against subsequent infections with that serotype alone. Subsequent secondary infections with another DENV serotype can result in fatal forms of DSS and DHF [25]. It is widely accepted that heterologous antibodies bind to DENV and, via its Fcγ receptor, increase uptake into monocyte-lineage cells, increasing viral load and causing severe dengue [26]. It is therefore essential that the DENV vaccine prevent infection by all four serotypes. The development of a dengue vaccine has been a complex process involving various vaccine prototypes with different attenuation properties, efficacies, and immunogenicity profiles. Some of the key points associated with the development of dengue vaccines are discussed in this review article.

## 2. Vaccine Prototypes

Several different vaccine models have been explored, such as live-attenuated chimeric recombinant virus, live-attenuated virus, inactivated virus, recombinant protein, and mRNA vaccines, as shown in Figure 2. The recently developed Sanofi Pasteur’s Dengvaxia is the first commercially licensed dengue vaccine consisting of a tetravalent formulation [27]. Despite being recommended and administered in multiple countries, Dengvaxia has shown severe adverse side effects and low effectiveness [28]. Ongoing efforts include the development of new strategies, such as the Takeda Biologics TAK-003 vaccine, which has been granted priority review by the U.S. FDA for the prevention of dengue disease caused by all four DENV serotypes. Moreover, challenges remain in dengue vaccine development, including the need for vaccines to induce equal levels of protection against all four DENV serotypes to avoid the risk of antibody-dependent enhancement (ADE) and the importance of considering the phenomenon of ADE in vaccine development.

## 3. Live Attenuated Vaccines

Live attenuated vaccines are made up of weakened forms of living pathogens that have been altered to be less harmful or non-pathogenic. These forms of vaccines offer the advantage of delivering a range of protective antigens and providing long-term immune protection. Various live dengue attenuated vaccines have been developed using recombinant DNA technology, including the chimeric yellow fever 17D (YF17D) virus–tetravalent dengue vaccine (CYD-TDV), the recombinant DENV-4 mutant with a 30-nucleotide deletion vaccine (rDEN4∆30), and the tetra-live attenuated virus dengue vaccine (DENVax) shown in Figure 3 [29]. Several vaccine formulations are at different stages of development at various organizations. So far, only one has been licensed for use, and several other vaccines are in clinical trials.

### 3.1. Dengvaxia

The CYD-TDV (chimeric yellow fever virus–DENV–tetravalent dengue vaccine), also known as Dengvaxia, is the first licensed dengue vaccine developed by Sanofi Pasteur. It has been approved in some countries, but it can only be used on people who have had dengue before. This tetravalent vaccine uses the YF17D vaccine strain as a backbone, substituting the YF17D prM and E regions with those of the four DENV serotypes [29]. For people aged 9–45 years, immunization is administered subcutaneously in 3 doses offered 6 months apart. The overall vaccine efficacy for symptomatic virologically confirmed dengue (VCD) was 60.3% in a pooled analysis of 25-month efficacy data from phase 3 trials in a population aged 2–16 years (95% CI: 55.7–64.5) [30]. However, the vaccine’s nonstructural proteins derived from the YF17D backbone have been reported to have limited effectiveness, which could make it less effective at triggering a protective T-cell response against DENV. Antibody viability was higher in older age groups, with 65.6% (95% CI: 60.7–69.9) in people between the ages of 9 and 16 compared to 44.6% (95% CI: 31.6–55.0) in children ages 2–8 [31]. Explicit immunization adequacy for DENV1 and DENV2 was 40% for both age groups, while it was 70–85% for DENV3 and DENV4. The immunization has a 72.7% viability (95% CI: 62.3–80.3) against hospitalization and an efficacy rate of 79.1% (95% CI: 60.0–89.0) against severe dengue across all age groups, taking serostatus into account. The relative risk of hospitalization in children between the ages of 2 and 9 is 1.58% higher (95% CI: 0.83–3.02) or 0.5% (95% CI: 0.29–0.86) for children between the ages of 9 and 16 [31]. Dengue risk may be affected by the patient’s baseline serostatus, and the vaccine may cause severe dengue in seronegative patients through ADE. Immunization viability is higher in seropositive people (70–80%) than in seronegative people (14.4–52.5%) in both age groups [32]. A subsequent report found that seronegative people have a higher risk of hospitalization and severe dengue, with a proportional hazard ratio (HR) of 1.75 for hospitalization and 2.87 for severe dengue, contrasted with corresponding HRs of 0.32 and 0.31 in seropositive patients [33]. The WHO suggests pre-immunization evaluation for nations considering CYD-TDV immunization, proposing inoculation of only seropositive people with confirmed past dengue infection or in regions with reported seroprevalence of approximately 80% by 9 years old.

### 3.2. QDENGA^®^ or TAK-003

Takeda Pharmaceuticals has developed a live attenuated, tetravalent dengue vaccine called QDENGA^®^ or TAK-003, which is based on chimeric viruses encoding dengue virus -2 vaccine strain 16,681 passaged in primary dog kidney cells (DEN-2 PDK-53) and tetra valent dengue vaccine 1–4 (TDV1–4) [34]. The vaccine has received approval for use in Indonesia for individuals aged 6 to 45 years, as well as marketing authorization in the European Union for people aged 4 years or older, regardless of their previous dengue infection history. In addition, TAK003 has been granted priority review by the U.S. Food and Drug Administration for the prevention of dengue disease caused by all four DENV serotypes [www.fda.gov/vaccines-blood-biologics/dengvaxia]. The vaccine is administered on a three-month schedule. Its approval is based on the Tetravalent Inoculation against Dengue Adequacy Study (TIDES), which involved over 28,000 participants. In the initial 11 months, the vaccine demonstrated 80.2% effectiveness in preventing dengue fever [35]. However, follow-up studies at 18, 24, and 36 months post-vaccination showed a cumulative efficacy of 62% against dengue fever [36]. Crucially, the data showed the vaccine provided consistent and robust protection against severe dengue, with effectiveness rates ranging from 83.6% to 90.4%. Importantly, the vaccine’s effectiveness was similar for those with and without prior dengue infection. While the vaccine was equally effective at producing specific antibodies across different serotypes, its effectiveness varied based on the dengue virus serotype. A new clinical trial (NCT03999996) is underway to investigate the impact of a booster dose given 15 and 24 months after the primary vaccination [37]. Early-stage studies also found that the vaccine stimulates specific immune responses against certain dengue virus proteins. Notably, the vaccine offers the highest protection against dengue virus type 2, from which the virus’s genetic material is derived, underscoring the importance of targeting both antibodies and T-cell responses for future dengue vaccine development.

### 3.3. TV003/TV005

This is a tetravalent vaccine developed by the National Institute of Allergy and Infectious Diseases (NIAID) and is currently in phase IIIB clinical trials as a live attenuated vaccine for dengue. It consists of three genetically attenuated viruses and one chimeric virus. This vaccine was created by incorporating 30 genetic deletions in the 3′ untranslated region and making further changes to non-primary proteins, resulting in weakened strains of the dengue virus—rDEN1Δ30, rDEN3Δ30/31, and rDEN4Δ30 [38,39]. These modified strains represent a version of the dengue virus in which 3′ UTR of the virus had specific mutations, leading to their attenuation. In two phase I clinical trials with placebo controls, a mild rash was the most frequently reported side effect, affecting around half of the participants in both the TV003 and TV005 groups [40]. However, a thorough analysis of the trial data revealed that a single dose of TV003 could achieve seroconversion rates of between 64% and 100%. Six months after administration of the attenuated DENV2 strain, rDEN230, TV003 effectively induced serotype-specific neutralizing antibodies, which offered protection against viremia, rash, and neutropenia.

The Butantan Institute in Brazil has been authorized to produce TV003 in a lyophilized form called Butantan-DV. Butantan DV, a single-dose tetravalent (four-strain) vaccine, showed 80% protection among participants with no evidence of previous dengue exposure and 89% protection in those with a history of exposure in phase III clinical trials [41]. In this study, the vaccine trial included results from 16 Brazilian centers located in all five regions of the country. At least 1 million Brazilians are infected with dengue each year; the current incidence rate in the country is 107.1 cases per 100,000 inhabitants, and the fatality rate is 0.9%. Two-year vaccine efficacy (VE) was 79.6% (95% confidence interval [CI]: 70.0–86.3) among participants with no evidence of previous dengue exposure and 89.2% (95% CI: 77.6–95.6) among those with a history of exposure. Overall, the four-strain VE against type 1 dengue was 96.8% and 85.6% among seropositive participants and seronegative participants, respectively, and 83.7% and 57.9%, respectively, against type 2 dengue. Dengue serotypes 3 and 4 were not circulating during the study period; hence, there was no assessment of efficacy against those strains.

## 4. Inactivated Virus Vaccines

Inactivated vaccines contain material derived from viruses that have been inactivated, allowing the immune system to recognize and respond to the viral components without the risk of causing disease. The immunogenicity of inactivated vaccines is assessed in animal models, and multiple booster doses are required to ensure long-term immunity, as they primarily express the structural proteins of the virus and may not stimulate a robust immune response against non-structural proteins. Several inactivated vaccine candidates in development and clinical trials are discussed below:

### 4.1. TDEV PIV

Inactivated vaccines contain pathogens (such as viruses or bacteria) that have been inactivated, allowing the immune system to recognize and elicit an immune response to the viral components without the risk of causing disease [42]. TDEV-PIV is a type of tetravalent inactivated dengue virus (DENV-1–4) vaccine, specifically a purified formalin-inactivated virus vaccine (TDENV-PIV) with alum adjuvant, created by the Walter Reed Army Institute of Research (WRAIR) [43]. TDEN-PIV contains all four serotypes of non-attenuated virus strains propagated in Vero cells, such as West Pac 74 (DENV-1), S16803 (DENV-2), CH53489 (DENV-3), and TVP360 (DENV-4) and inactivated with formalin [43]. In Phase I trials, TDEV-PIV showed general variation but was notably consistent across the groups in inducing T-cell responses mediated by IFN-γ. However, each group encompassed a range of responses, from individuals with exceptionally high levels (over 1000 SFC/106 PBMC) of responses to all serotypes to others with minimal responses (below 38 SFC/106 PBMC) against any serotype. The DENV-2 serotype was identified as the most common, eliciting the most significant IFN-γ response across all groups at both time points [44]. The study compared the immune response in Macaca mulatta (rhesus monkeys) that received either the tetravalent protein (TPIV) or tetravalent DNA (TDNA) vaccine and was subsequently boosted with a tetravalent live attenuated vaccine (TLAV) [45]. It was found that those vaccinated with both types of vaccines demonstrated significantly higher levels of humoral immunity against Dengue viruses (DENVs) compared to those vaccinated with only one type of vaccine.

### 4.2. DENV-2 Vaccine S16803

The DENV-2 vaccine S16803 was also formulated by the Walter Reed Army Institute of Research (WRAIR), who grew the virus in Vero cells and used formalin inactivation and sucrose-centrifugal processes [42,46]. It was also tested with four different adjuvants and demonstrated effectiveness in animal models, such as rhesus monkeys, through formalin inactivation and purification methods [47]. Putnak et al. showed the differences in the immunogenicity of the two vaccines, inactivated vaccine S16803, with that of the recombinant subunit protein vaccine (R80E) and LAV (DENV2 PDK-50). Their results showed that DENV2 PDK-50 produced a more stable titer of antibodies [46]. However, these vaccines typically require multiple booster doses to ensure long-term immunity, as they primarily express the structural proteins of the virus and may not stimulate a robust immune response against non-structural proteins. While inactivated vaccines have the advantage of being stable and less likely to cause pathogenic effects, they also face challenges. These include the need for adjuvants to enhance immunogenicity, which can increase costs and potential reactogenicity. Additionally, ensuring that the vaccine induces both humoral and cell-mediated immune responses is crucial for effective protection. Finally, inactivated viral vaccines require careful consideration of their formulation and the immune responses they elicit to be effective.

## 5. Subunit Vaccines

Subunit vaccines are constructed using defined protein antigens, particularly the envelope (E) proteins, produced in heterologous expression systems that act as antigens to stimulate an immune response. These vaccines can be produced using various expression systems, including eukaryotic and prokaryotic cells, to ensure the generation of effective immune responses.

### 5.1. EDIII-P64K

Recent studies have shown that recombinant envelope protein domain III (EDIII), derived from E. coli, can effectively induce the production of antibodies against all dengue serotypes in animal models [48]. This indicates the potential of subunit vaccines to provide broad protection. The tetravalent dengue vaccine EDIII-P64K combines the EDIII region of different DENV serotypes and the P64K protein of Neisseria meningitidis [49]. This vaccine has demonstrated the ability to produce high levels of antibodies against DENV serotypes in mice and monkeys [48,50]. DENV E domain III-P64k recombinant proteins showed high antigenic specificity and a low potential for inducing cross-reactive antibodies in immunized mice and monkeys [51]. Subunit vaccines offer several advantages, including a reduced risk of adverse effects compared to live attenuated vaccines, as they do not contain live virus. Moreover, the expression of a fusion between DENV1–2 and DENV3–4 EDIII, coupled with a Gly-Ser linker, was expressed in Escherichia coli [52]. This fusion construct was found to effectively stimulate immune protection against the four serotypes of dengue virus in mice.

### 5.2. V180 (DEN-80E)

V180, also known as DEN-80E, is a tetravalent subunit vaccine co-developed by Merck and Medigen Vaccine Biologics (MVB). It consists of a truncated recombinant protein that captures 80% of the envelope protein (E) derived from strains of all four dengue virus serotypes (DEN-1 strain 258848, DEN-2 strain PR159 S1, DEN-3 strain CH53489, and DEN-4 strain H241) [53]. The DEN-80E subunits are produced within the drosophila S2 cell expression system using plasmids and then combined with either ISCOMATRIX, a formulation consisting of saponin, cholesterol, and phospholipid adjuvant (CSL), or Alhydrogel [53]. The V180 vaccine is generally considered safe due to the lack of live virus components. Its inability to replicate within the host may also limit the duration of immune responses, potentially necessitating additional doses for long-term immunity. Currently, V180 is part of ongoing research and clinical trials to assess its safety and efficacy in preventing dengue virus infections. In phase III clinical trials, the V180 vaccine candidate showed a 79.6% effectiveness rate in preventing dengue. During this trial, no serious cases of dengue were reported among the participants, suggesting that subunit vaccines are safer, including having a reduced risk of adverse effects compared to live attenuated vaccines, as they do not contain live virus [54]. This allows the vaccine to stimulate a strong and balanced immune response against all four dengue virus serotypes while minimizing the risk of antibody-dependent enhancement (ADE) [54], which is crucial for the effectiveness of vaccination as it helps to reduce the severity of dengue disease associated with heterotypic infections. However, subunit vaccines also face several challenges, such as the need for adjuvants to enhance immunogenicity and the potential for improper protein folding, which can affect the vaccine’s efficacy. Subunit vaccines represent a promising strategy in dengue vaccine development, with ongoing research aimed at optimizing their formulation and enhancing their immunogenicity to provide effective protection against dengue virus infections.

## 6. Viral Vector Vaccines

Viral vector vaccines employ viruses that have been genetically modified to be replication-defective and cannot cause disease in humans but capitalize on their natural ability to enhance immunostimulatory properties. The immunostimulatory effects of these viruses facilitate the activation of innate immunity, thereby enhancing the response to the target antigen. The viral vector acts as a delivery system, introducing the genetic material encoding the dengue antigens into host cells. These vaccines use various viral vectors. Common viral vectors used in dengue vaccine development include adenoviruses, alphaviruses, and the Vaccinia virus [8]. These vectors are chosen for their ability to induce strong immune responses and their capacity for genetic manipulation. Adenoviral vectors are noted for their ease of genetic manipulation, allowing for the efficient expression of dengue antigens. A recombinant replication-defective adenovirus vector expressing dengue envelope proteins has been shown to elicit robust antibody responses in animal models [55]. This is particularly beneficial for diseases like dengue, where protection depends on a robust immune response. A recombinant replication-defective adenovirus vector expressing the genes encoding prM and E proteins of dengue virus types 1 and 2 (CAdVax-Den12) or dengue virus types 3 and 4 (CAdVax-Den34) was produced [56]. Administration of these two divalent vaccines combined with intramuscular inoculation in rhesus monkeys resulted in a significant antibody response against all four dengue virus serotypes and effectively neutralized all four dengue virus serotypes in vitro [57]. On the other hand, a replication-deficient modified vaccinia Ankara (MVA) vector was used to express 80% of the full-length, highly immunogenic C-terminally truncated E protein of either DENV2 or DENV4 viruses. The efficacy of these vectors in conferring protective immunity was assessed in animal models. The results showed that the E protein of DENV2 demonstrated significantly higher levels of anti-E antibody titers compared to DENV4. However, further research showed that a three-dose schedule utilizing the E protein of DENV2 resulted in an increased antibody response and enhanced protection against DENV2 [58]. Another study combined the DIII domains of the four serotypes into a single amino acid sequence using flexible 2x(GGGS) linkers; the resulting protein was named d34 and expressed in an MVA viral vector. MVA-based vaccines induce a good T-cell response, which may be important for protection against flaviviruses. The sera of vaccinated mice show virus-neutralizing activity against dengue serotype 2. Thus, the developed MVA-d34 vaccine is a promising candidate vaccine against dengue fever [59]. In addition, Venezuelan equine encephalitis virus replicon particles (VRP) have been explored as a platform for dengue vaccines, expressing two types of dengue virus envelope antigens (soluble E dimers [E85] and sub-viral particle [prME]) demonstrating the potential to induce protective immunity in animal models [60].

In addition, baculovirus-expressed NS1 protein, used as a vaccine candidate [61], provided only partial or low levels of protection in mice when exposed to DENV challenges. The DENV-2 NS1 recombinant protein is expressed in baculovirus-infected insect cells, which were found to be similar to the authentic NS1 of DENV-infected Aedes albopictus cells in terms of glycosylation, dimerization, cellular presentation, and antigenicity. Mice administered the baculovirus-expressed NS1 protein developed NS1-specific complement-fixing antibodies, which partially protected them against neurological residuals following DENV2 infection via intracellular challenge. Further investigations into the efficacy of the baculovirus–DENV-4 NS1 recombinant protein in Rhesus macaques immunized with cell extracts from these recombinant proteins, including DENV4 structural and non-structural proteins (C-M-E-NS1-NS2a), revealed that most of the monkeys did not acquire protection [62]. Overall, viral vector vaccines offer several benefits, including the ability to induce both humoral and cellular immune responses, ease of genetic modification, and high levels of protein expression. They can also provide a balanced immune response, which is crucial for effective protection against the different dengue serotypes. However, one of the challenges with viral-vectored vaccines is that individuals may have pre-existing immunity to the viral vector used, which can reduce the vaccine’s effectiveness. This pre-existing immunity can lead to an immune response against the vector rather than the target antigen. Some challenges have been encountered when expressing certain dengue virus proteins in specific viral vectors, which can affect the overall efficacy of the vaccine. Research is ongoing to develop effective viral-vectored vaccines for dengue.

## 7. DNA Vaccines

DNA vaccines consist of plasmids that encode specific antigens from the dengue virus. When these plasmids are injected into a host, the cells take them up and then produce the encoded antigens. This process leads to the stimulation of both humoral and cellular immune responses. Research has demonstrated that DNA vaccines can effectively trigger the production of neutralizing antibodies in animal models.

### 7.1. TVDV

This is a tetravalent DNA vaccine based on prM and E protein-coding sequences that is co-administered with VAXFECTIN as an adjuvant. It was developed by U.S. AMRDCh, WRAIR, NMRC, and Vical and is currently in animal and phase I evaluation. Administration of TVDV with adjuvant was found to effectively stimulate a strong T-cell IFNγ response against dengue, showing a better safety profile [63]. To assess efficacy, a combination of tetravalent DNA vaccine (TVDV), tetravalent purified formalin-inactivated virus (TPIV), and tetra-live attenuated virus (TLAV)-enhanced vaccination strategy was used in rhesus monkeys. Monkeys immunized with TVDV/TVDV/TLAV were partially protected, while those immunized with TPIV/TLAV were completely free of viremia [64]. Additionally, data from phase 1 clinical trials revealed that TVDV with Vaxfectin^®^ adjuvant triggers an anti-dengue T-cell IFNγ response with optimal safety [63].

### 7.2. D1ME100

This DNA vaccine expresses specific dengue virus antigens, particularly prM and 92% of the envelope (E) genes, which are crucial for inducing an immune response [65,66]. This vaccine aims to induce both humoral (antibody-mediated) and cellular immunity against the dengue virus. D1ME100 can effectively induce anti-dengue antibodies in animal models [65]. An intradermal vaccination of mice with D1ME100 has been reported to produce a strong antibody response, indicating its potential effectiveness as a vaccine. Despite its potential, D1ME100, like other DNA vaccines, faces challenges in achieving high immunogenicity. Strategies to enhance its effectiveness may include using highly efficient promoters, exploring alternative delivery methods, co-immunization with adjuvants, and incorporating immunostimulatory motifs.

### 7.3. DENV EDIII-Based DNA Vaccine (DDV)

The DENV domain III of E protein (EDIII)-based DNA vaccine (DDV) candidate is created by combining the consensus sequence from the E protein domain III (EDIII) of dengue virus serotypes 1–4 and a dengue virus (DENV)-2 non-structural protein 1 (NS1) protein-coding region, instead of a single-strain genome sequence. Recent studies have shown that EDIII-based DENV vaccines evade ADE of infection in mice, whereas T-cell response against the EDIII vaccine plays a role in disease protection via effective viral clearance [67]. NS1 has been shown to activate T-cell responses involved in protection against dengue infection in both humans and experimental animals. Additionally, the T-cell response to the EDIII DNA vaccine has been demonstrated to efficiently eliminate the virus. Mutations in the EDIII region may impact both antibody binding and the protein’s interaction with cellular receptors [68]. Taken together, the EDI and NS1 vaccine designs could reduce immune interference among serotypes.

Additionally, DNA vaccines can induce both T-cell and antibody responses, which are crucial for effective immunity against dengue. A study by Porter et al. showed that a DNA vaccine expressing the prM and E proteins of DENV-2, along with immunostimulatory motifs, could enhance the immune response [69]. Moreover, during DNA vaccine development, certain guiding proteins are used to increase the immune system’s response; for instance, the antigen sequence integrated into the lysosomal membrane protein, leading to increased expression of MHC class II antigens. This increased expression enhances the production of CD4 T-cells and anti-CD4 antigens, ultimately increasing the immunogenicity of the DENV2 prM/E DNA vaccine [56]. This suggests that a DNA vaccine expressing the prM and E proteins of DENV-2, along with immunostimulatory motifs, could enhance the immune response. Furthermore, combining DNA vaccines with other vaccine types, such as viral vector vaccines, has shown promise in improving immunogenicity and protection. Additionally, a DNA vaccine that includes an antigen fused with a single-chain Fv antibody (scFv) specific for the DC endocytic receptor could trigger a robust immune response targeted at the antigen. Moreover, combining DNA vaccines with other vaccine types, such as viral vector vaccines, has shown promise in improving immunogenicity and protection. In addition, DNA vaccines offer several advantages, including stability, cost-effectiveness, and suitability for mass production. They are non-replicating and non-infectious, which reduces the risk of causing disease in vaccinated individuals. However, DNA vaccines also face challenges related to their immunogenicity. Strategies to enhance their effectiveness include using alternative delivery methods, promising promoters, and co-administration with adjuvants.

## 8. Virus-like Particle (VLP) Vaccines

Virus-like particle (VLP)-based vaccines mimic the structure of viruses without containing their genetic material. These particles are recognized by antigen-presenting cells (APC), facilitating receptor-mediated uptake and subsequent presentation to lymphocytes, thus initiating a target-specific immune response. VLPs can be produced from several sources, such as bacteria, yeast, insects, mammals, and cell-free expression systems.

The DENV1–4 VLP vaccine was created by co-expressing the precursor membrane (prM) and envelope (E) proteins of the four dengue virus serotypes [70]. The E protein’s fusion loop was mutated to enhance VLP production in mammalian cells. The optimized DENV1–4 VLPs were expressed in 293F cells and purified from the culture supernatants [70]. The purified DENV1–4 VLP samples contained the E (53 kDa) and prM (around 20 kDa) proteins as the major components. Recent non-human primate studies have demonstrated that the DENV VLP (DENVLP) vaccine generates a robust neutralizing antibody response against all four DENV serotypes, which is maintained for up to 1 year. Importantly, the DENVLP vaccination did not elicit any antibody-dependent enhancement (ADE) response against any of the four DENV serotypes in vitro. Furthermore, the DENVLP vaccine was able to reduce viral replication in a non-human primate challenge model. Passive transfer of purified IgG from immunized monkeys into immunodeficient mice provided protection against subsequent lethal DENV challenge, indicating a humoral mechanism of protection.

Immunization of non-human primates with a tetravalent DENVLP vaccine induced high levels of neutralizing antibodies and reduced the severity of infection with all four dengue serotypes. VLPs represent a promising platform for future vaccine development, as they are safe and can be produced in large quantities using various expression systems, making them a cost-effective option for vaccine production. However, the development of VLP vaccines faces challenges in ensuring the correct assembly of the particles and achieving optimal immunogenicity. VLPs made from bacteria and insects may have impurities like endotoxin or baculovirus, and a lack of post-translational protein modifications can lead to formulation stability issues. Ongoing research is focused on optimizing the formulation and enhancing the immunogenicity of VLP vaccines to provide effective protection against dengue virus infections, exploring the use of various adjuvants and delivery methods.

## 9. Dengue mRNA Vaccine

mRNA vaccines work by introducing modified messenger RNA (mRNA) into the body, which encodes specific dengue virus proteins. Once inside the cells, this mRNA is translated into viral proteins, prompting an immune response. The use of lipid nanoparticles (LNPs) to deliver the mRNA enhances its stability and uptake by cells. The authors of [71] developed an mRNA vaccine targeting the DENV-1 membrane and envelope structural proteins (prM/E). This vaccine was created using pseudo-uridine-modified mRNA, which was then encapsulated in lipid nanoparticles for intramuscular injection. The resulting immune response in study mice was serotype-specific, indicating the potential for effective immunization.

Zhang et al. studied mRNA vaccines by targeting additional proteins of DENV-2, such as prME, E80, and NS1, using a similar mRNA technology [72]. This candidate vaccine has shown the ability to induce neutralizing antibodies against DENV-2 and trigger a T-cell immune response in mice. All four serotypes showed high serum DENV antibody titers; when the cytokine stages were analyzed, the results showed that IgG2a production in response to Th1 cytokine IFN-γ was much higher than IgG1 production in response to Th2 cytokine IL-4, suggesting that the mRNA vaccine-mediated Th1 response is dominant. In addition, mRNA vaccines offer several advantages, including rapid development and the ability to elicit strong immune responses. The flexibility of mRNA technology allows for quick modifications to target different serotypes of the dengue virus, which is crucial given the existence of multiple serotypes (DENV-1 to DENV-4). However, the major concern is that the E protein mutated with E-DII does not effectively mitigate the effects of antibody-dependent enhancement (ADE) within the mRNA vaccine framework, which leads to insufficient immunogenicity [71] (Table 1). It is also important to study the ADE effects associated with immune serum targeting either the NS1 or E-DIII proteins individually and to conduct a comparative analysis and assessment of polyvalent vaccines. Further investigation in this area is needed.

## 10. Summary

The pathogenesis of dengue is intricate, involving multiple factors from both the host and the virus. A thorough understanding of these complex interactions is essential for the development of effective vaccines. Researchers highlight the necessity of understanding how each dengue virus serotype contributes to the severity of infection and the effectiveness of the vaccine (summarized in Table 1). They stress the importance of creating a vaccine that can offer comprehensive protection against all four serotypes, while ensuring it is safe and effective for widespread use. The development of dengue vaccines faces several significant challenges. The FDA-approved vaccine “Dengvaxia” is effective only in individuals aged 9 to 16 years who have previously contracted dengue. However, it carries a risk of severe dengue in seronegative individuals, especially in children under 9, resulting in higher hospitalization rates. This risk of antibody-dependent enhancement (ADE) presents a critical safety issue that needs to be addressed. Antibody-dependent enhancement (ADE) occurs when non-neutralizing antibodies allow a virus to enter host cells, increasing viral load. Studies indicate that antibodies targeting the envelope protein domain III (EDI) may provide effective neutralization while minimizing the risk of ADE. Conversely, antibodies targeting the pre-membrane (prM) and fusion loop epitope (FLE) regions have been associated with an increased risk of ADE. A thorough understanding of the structural and molecular mechanisms underlying both antibody-mediated neutralization and ADE is essential for developing safe and effective dengue vaccines.

The primary challenges in creating a successful dengue vaccine include the following: -Achieving broad protection against all four dengue virus serotypes to reduce the risk of ADE.-Addressing the issue of ADE, where prior exposure to one serotype can lead to more severe disease upon subsequent infection with a different serotype.-Ensuring balanced immunogenicity, as some vaccine candidates (such as DNA and subunit vaccines) may need adjuvants to boost their effectiveness, complicating development and increasing costs.-Addressing safety concerns, including the potential for heightened disease severity in vaccinated individuals.-Producing vaccines, particularly subunit and inactivated types, in a cost-effective and logistically viable manner, as the dengue virus’s inability to grow to high titers in tissue culture cells complicates the production of inactivated vaccines.

Despite these difficulties, the NIAID LATV TV003/TV005 vaccine emerges as a leading candidate based on available clinical trial data. It demonstrates robust neutralizing antibody responses, successful seroconversion for all four serotypes, and the benefit of a single-dose regimen, making it a viable option for implementation in developing countries. Ongoing research and innovative approaches are crucial to overcome these challenges and develop safe, effective dengue vaccines for widespread use in endemic regions. In conclusion, the document highlights the ongoing efforts and advancements in Dengue vaccine development, emphasizing the importance of addressing the challenges and seizing the opportunities to create an effective vaccine that can combat the global threat of dengue virus.

## 11. Future Directions in Dengue Vaccines

There is an urgent need for a dengue vaccine that provides long-term protection against all four dengue virus serotypes. Several promising dengue vaccine candidates are currently in preclinical and clinical stages, and there is growing optimism about the future of a dengue vaccine. Innovations in vaccine technology, ongoing clinical trials, and the search for new platforms raise expectations for developing an effective and safe dengue vaccine. However, challenges remain in vaccine development, including variations in immune responses to different serotypes, ensuring long-term and balanced protection, and addressing safety concerns, especially for individuals who have not been previously exposed to dengue. Early preparation and a clear understanding of the actual burden of the disease will be essential for the successful introduction of the vaccine. It is crucial to maintain adequate surveillance to assess the safety and effectiveness of dengue vaccines during the post-licensure period. In addition to conventional approaches, unique DENV multiplication and lifecycle aspects may provide focal points for innovative vaccine formulations. Most dengue vaccines target either the viral structural protein NS1 or the envelope proteins present on the virus surface, yet there are still many safety concerns that must be addressed. In addition to the NS1 protein, other non-structural proteins such as NS2A, NS2B, NS3, NS4A, NS4B, and NS5 also present avenues for inhibitor development.

Several studies have explored the link between human leukocyte antigen (HLA) genotypes and both susceptibility to and severity of dengue infections. Multiple HLA alleles have been identified as factors of susceptibility or protection against dengue across diverse populations [73,74], including HLA class I and II alleles and proteins like MICA, MICB, and LTA. HLAs are critical in mediating both innate and adaptive immune responses during viral infections. Infection of primary monocytes with the dengue virus (DENV) leads to upregulated HLA class I molecules during acute illness, which suppresses natural killer cell activity [75]. Dengue virus consistently induces strong CD8+ T-cell responses, whereas specific HLA class I variations are significant risk factors for dengue hemorrhagic fever. Current efforts focus on developing multi-epitope peptide vaccines that include B-cell, CD8+, and CD4+ T-cell epitopes targeting specific HLA types to enhance efficacy [76]. Recent research has focused on HLA-restricted T-cell epitopes for vaccine development, leading to the identification of novel MHC-I-restricted epitopes, HLA-A2 and HLA-A24, which activate CD8+ T-cell responses [77]. Genome-wide analyses and advancements of computational methods in the field of vaccine development have identified key HLA factors associated with severe dengue infection across various scenarios [78] and conserved immunogenic epitopes suitable for population-wide vaccine development [79]. These findings are crucial for advancing effective multi-serotype dengue vaccines.

Additionally, mRNA vaccines could prove to be a promising avenue, especially given the success of the mRNA COVID-19 vaccines developed during the pandemic. One main goal of dengue vaccine research is to develop a universal vaccine that offers long-lasting protection against all four DENV serotypes. Current candidates typically target single serotypes, but a universal approach could simplify vaccination and reduce incomplete protection risks. An effective DENV vaccine should also target the divergence of circulating strains; thus, a consensus-based vaccine may be optimal. Such vaccines could minimize sequence diversity across strains. Recent studies on H1N1 and HIV in mice have suggested that consensus immunogen-based vaccines may be crucial in addressing viral genetic divergence. Furthermore, several studies have indicated that consensus prediction may outperform single-sequence determination methods in vaccine design.

New vaccine platforms like mRNA, viral vectors, and nanoparticles show promise in enhancing immunogenicity and optimizing immune responses. Additionally, exploring combination vaccines that address multiple diseases, such as Zika and Chikungunya, could increase efficacy and streamline prevention strategies. Ongoing research aims to improve the safety and effectiveness of dengue vaccines, particularly for individuals with no prior dengue exposure. In addition, computational advances in immuno-informatics can lead to better vaccine designs by improving the identification of antigenic sites and predicting viral evolution. While research is ongoing to develop inhibitors for various NS proteins, none have yet reached clinical trials due to challenges in structural studies.

## Figures and Tables

**Figure 1 viruses-17-00212-f001:**
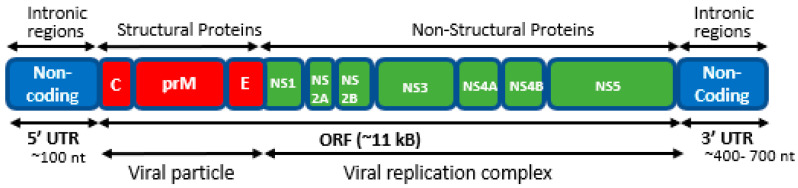
Genome structure of dengue virus. The genome of the dengue virus comprises both structural and non-structural proteins. Non-coding regions have also been shown. C: capsid; prM: pre-membrane; E: envelope; NS: non-structural proteins; UTR: untranslated regions; ORF: open reading frame.

**Figure 2 viruses-17-00212-f002:**
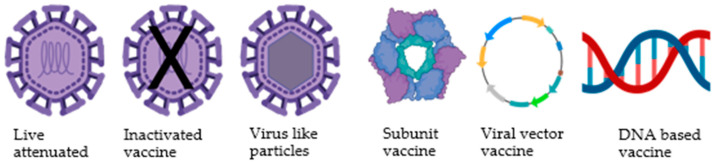
Types of vaccine prototypes under development. Different types of dengue vaccine formulations are being developed by various organizations. (VLP and mRNA-based vaccines were not depicted here). Image created with Biorender.com.

**Figure 3 viruses-17-00212-f003:**
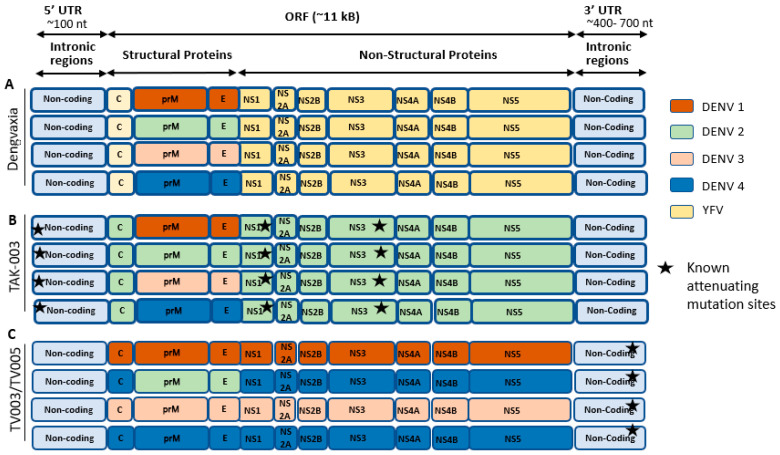
Types of live attenuated tetravalent dengue vaccines. The schematic represents live attenuated tetravalent dengue vaccine formulations. (**A**) Dengue genome components of Sanofi’s Dengvaxia consist of the YFV-17D virus backbone (indicated in yellow) to facilitate the expression of the prM and E genes from DENV1, DENV2, DENV3, and DENV4. (**B**) TAK-003 contains an attenuated DENV2 strain backbone (green) and three chimeric virus strains expressing the prM and E genes of DENV1 (brown), DENV3 (pink), and DENV4 (blue). (**C**) NIH/ Bhutantan (TV003/TV005) TV003/TV005 comprises three complete DENV strains that include the entire DENV genome (both structural and non-structural genes) along with one chimeric virus, where the prM and E genes of DENV4 are replaced with those of DENV2. These viruses are rendered attenuated by a common deletion of either 30 nucleotides (Δ30) or 31 nucleotides (Δ31) in the 3′ UTR of the viral genome. C: capsid; prM: pre-membrane; E: envelope; NS: non-structural proteins.

**Table 1 viruses-17-00212-t001:** Summary of dengue vaccines under development.

Vaccines	Nature	Strategy and Target	Strength	Weakness
Dengvaxia^®^ (CYD-TDV)	Attenuated	YF-17D backbone with prM and E genes of dengue virus 1–4	Immune response against all four serotypes. (Licensed)	High risk of dengue-related hospitalization of children below 9 years of age. Low vaccine efficacy to DENV-2.
QDENGA^®^ or TAK-003	Attenuated	DENV2 PDK53 backbone with DENV1/3/4 prM and E gene chimera	A single vaccine dose with high efficacy rates and no serious adverse events reported (Licensed)	Vaccine efficacy data are not available for individuals > 16 years old
TV003/TV005	Attenuated	Full length DENV1,2,3,4 lacking 30 nucleotides in 3′ UTR	A single dose of TV-005 produced a tetravalent response in 90% of the vaccinated (phase III trial)	No data available
TDEV-PIV	Inactivated	Tetravalent formalin-inactivated virus (TPIV)	Immune response against all four serotypes (broad protection)	Lower immunogenicity. Adjuvants needed
DENV-2 vaccine S16803	Inactivated	Formalin-inactivated virus (DPIV)	Stable and less pathogenic; Vero cell generated	Need multiple booster injections. Less immunogenicity.
EDIII-P64K	Subunit	Recombinant EDIII derived from *E. coli*	Immune response against all four serotypes (broad protection)	Not tested in humans
V180 (DEN-80E)	Subunit	DENV recombinant truncated 80 E	Low cross-reactive antibodies. Insect cell generated (phase III trials)	Adjuvants are needed, and improper protein folding causes concerns.
TVDV (Vaxfectin)	DNA	Recombinant plasmid vector encoding prM/E proteins of DENV1–4	Potent neutralizing antibodies against all four serotypes (phase 1 trial)	Protein misfolding and exposure to endotoxins in plasmid preparation remain.
DIME100	DNA	Recombinant plasmid vector encoding prM/E	Safe and well tolerated (phase 1 trial)	Lower immunogenicity. No neutralizing antibody response.
DENV EDIII-based (DDV)	DNA	DENV2-EDIII	Safe and well tolerated (phase 1 trial)	Lower immunogenicity. No neutralizing antibody response.
DENVLP	Virus-like particle (VLP)	prM and E protein-coding sequences	Immune responses against all four serotypes.	Impurities like endotoxin or baculovirus. Formulation stability issues
prME-mRNA, E80-mRNA, and NS1-mRNA	mRNA	Consensus sequences of EDIII of DENV1–4 and DENV-2 NS1, and prME, E80, and NS1 of DENV-2	Induce the production of neutralizing antibodies against DENV-2	Relatively new technology and long-term effects are unknown.

## Data Availability

Not applicable.

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
