# Peer review of "Current Dengue Virus Vaccine Developments and Future Directions"

_viruses, 2025, doi:10.3390/v17020212_

Round 1
Reviewer 1 Report
Comments and Suggestions for Authors
The authors provided a comprehensive review of the current state of dengue virus vaccine development, highlighting various vaccine types, including live attenuated, inactivated, and DNA vaccines, while addressing challenges such as achieving broad protection against all four dengue serotypes and the risks associated with antibody-dependent enhancement (ADE). It is a well organized review. I have only one concern that needs improvement: ADE is very important for the immunology and vaccines of dengue virus. Some studies show that the target of ADE antibodies on the E protein may overlap with the neutralizing epitopes, which is a huge challenge for vaccine development. I hope the authors can discuss more about ADE and the development of dengue vaccines.
Author Response
We appreciate all the reviewers' time and efforts in constructively criticizing the review, entitled “Current Dengue Virus Vaccine Developments and Future Directions,” submitted for publication in the Journal Viruses. Please see below our responses to each comment made by the individual reviewers.
Reviewer 1:
Comments 1: Some studies show that the target of ADE antibodies on the E protein may overlap with the neutralizing epitopes, which is a huge challenge for vaccine development. I hope the authors can discuss more about ADE and the development of dengue vaccines
Response: I agree with reviewers that ADE is a critical bottleneck for Dengue vaccine development. Throughout the manuscript, we have discussed ADE wherever it is needed, based on the type of vaccine discussed. Additionally, we added a few more sentences to the summary.
Reviewer 2 Report
Comments and Suggestions for Authors
This review details all the technologies and platforms that have been used to create vaccines against dengue virus. The review includes both historical examples of vaccine development and the most recent ones. The advantages and disadvantages of all the vaccines described are described in detail. Future prospects for the development of new vaccines are detailed. The review is very informative and is recommended for acceptance after corrections by the authors.
Major revisions:
In the figure 2 it is necessary to give the real scheme of flaviviruses, not viruses from other families (probably Coronavirus is given) you can use, for example, Cryo-Em structure from the link and another flavivirus (doi: 10.1038/s41467-018-02882-0).
Line 205-207 - It's not clear what's meant. «These modified strains represent a version of the dengue virus in which the prM and E proteins in rDEN4Δ30 206 have been substituted with those from the dengue virus.»
Minor revisions:
Line 107 - use the abbreviation Fcɣ for Fc-gamma receptor instead of Fcg
Line 108 - missing dot at the end of the sentence
Line 182 – incorrect formatting
Line 200 – Missing first word in paragraph
Fig.3 - Bring the color scheme of Dengue 1 to a consistent color scheme
Line 365 - No capitalization
Line 411 – Missing first word in paragraph
It is necessary to harmonise the reference list with the same style
Author Response
We appreciate all the reviewers' time and efforts in constructively criticizing the review, entitled “Current Dengue Virus Vaccine Developments and Future Directions,” submitted for publication in the Journal Viruses. Please see below our responses to each comment made by the individual reviewers.
Major revisions:
Comment 1: In the figure 2 it is necessary to give the real scheme of flaviviruses, not viruses from other families (probably Coronavirus is given) you can use, for example, Cryo-Em structure from the link and another flavivirus (doi: 10.1038/s41467-018-02882-0).
Response: The reviewer is correct that the previous picture appeared to be a coronavirus, which was an accident. The idea was to take a virus prototype, and a coronavirus-looking diagram was chosen. We changed the diagram altogether. We created a picture using Biorender.com, which is unique for this purpose.
Comment 2: Line 205-207 - It's not clear what's meant. «These modified strains represent a version of the dengue virus in which the prM and E proteins in rDEN4Δ30 206 have been substituted with those from the dengue virus.»
Response: The previous sentence was changed to the following to clarify. “These modified strains represent a version of the dengue virus in which 3` UTR of the virus had specific mutation leading to their attenuation.”
Minor revisions:
Comment 3:
Line 107 - use the abbreviation Fcɣ for Fc-gamma receptor instead of Fcg
Response: Corrected
Line 108 - missing dot at the end of the sentence
Response: Corrected
Line 182 – incorrect formatting
Response: Corrected
Line 200 – Missing first word in paragraph
Response: Corrected
Fig.3 - Bring the color scheme of Dengue 1 to a consistent color scheme
Response: Corrected
Line 365 - No capitalization
Response: Corrected
Line 411 – Missing first word in paragraph
Response: Corrected
It is necessary to harmonise the reference list with the same style
Response: We agree with the reviewer and made essential corrections throughout the article reference.
Reviewer 3 Report
Comments and Suggestions for Authors
This manuscript provides a comprehensive overview of the current developments and future research directions in dengue vaccine research. It elaborates on various vaccine development approaches, including attenuated, inactivated, subunit, and viral vector vaccines. Additionally, the manuscript leverages preclinical and clinical research data to analyze the strengths and limitations of each vaccine type, aiming to contribute to the improvement of public health issues related to dengue fever. The manuscript is logically structured, thoroughly comprehensive, and integrates current research hotspots, with a focus on future research directions. However, there are still numerous issues in the manuscript, particularly in terms of writing format and content, which require substantial revisions. Therefore, major revision has to be done before this manuscript could be accepted. Below are some review comments provided to help further improve the manuscript.
1. The manuscript discusses the advantages and disadvantages of each vaccine but lacks a systematic and comprehensive comparative analysis. It is recommended to add a table that briefly and concisely highlights the strengths and weaknesses of different vaccines, making the content more intuitive.
2. There is a lack of discussion regarding the limitations of this manuscript.
3. Some of the data sources used in the article are outdated, such as the experimental data for Dengvaxia on page 4. It is recommended to utilize the most recent data to ensure timeliness.
4. It is recommended to use more recent references, and it is advised not to cite too many sources from around the year 2000.
5. Figure 2, Figure 3, and Table 1 are not cited in the text. It is necessary to reference them appropriately in the relevant sections of the manuscript.
6. In Figure 1 and Figure 3, abbreviations should be clearly defined with their full terms.
7. There are numerous language and formatting errors, thorough proofreading and revision are recommended.For example:①Line 108: There is no punctuation mark following reference [26].②Line 200: The first letter should be capitalized.③Line 492: The reference citation placement is incorrect, and there is a grammatical error.④Line 558: There is a punctuation error.⑤The number of spaces after punctuation marks throughout the manuscript is inconsistent.
Comments on the Quality of English LanguageThe English could be improved to more clearly express the research.
Author Response
Comment 1: The manuscript discusses the advantages and disadvantages of each vaccine but lacks a systematic and comprehensive comparative analysis. It is recommended to add a table that briefly and concisely highlights the strengths and weaknesses of different vaccines, making the content more intuitive.
Response: The table # 1 has been redesigned to accommodate the advantages and disadvantages of each vaccine.
Comment 2: There is a lack of discussion regarding the limitations of this manuscript.
Response: We added the advantages and disadvantages of different vaccines against the Dengue virus.
Comment 3: Some of the data sources used in the article are outdated, such as the experimental data for Dengvaxia on page 4. It is recommended to utilize the most recent data to ensure timeliness.
Response: We are citing the original manuscripts. These vaccines were developed and tested long ago, and thus, the manuscripts describing them are old.
Comment 4. It is recommended to use more recent references, and it is advised not to cite too many sources from around the year 2000.
Response: Some original vaccine trials are old, and we tried to cite the original work. However, on some occasions, we used old citations, which were replaced with new ones.
Comment 5. Figure 2, Figure 3, and Table 1 are not cited in the text. It is necessary to reference them appropriately in the relevant sections of the manuscript.
Response: It has been corrected.
Comment 6. In Figure 1 and Figure 3, abbreviations should be clearly defined with their full terms.
Response: It has been corrected.
Comment 7. There are numerous language and formatting errors, thorough proofreading and revision are recommended.For example:①Line 108: There is no punctuation mark following reference [26].②Line 200: The first letter should be capitalized.③Line 492: The reference citation placement is incorrect, and there is a grammatical error.④Line 558: There is a punctuation error.⑤The number of spaces after punctuation marks throughout the manuscript is inconsistent.
Response: It has been corrected.